# Noninvasive optical activation of Flp recombinase for genetic manipulation in deep mouse brain regions

Hyunjin Jung[1], Seong-Wook Kim[2], Minsoo Kim[2], Jongryul Hong[1], Daseuli Yu[1], Ji Hye Kim[2], Yunju Lee[2], Sungsoo Kim[1], Doyeon Woo[2], Hee-Sup Shin[2], Byung Ouk Park[2] & Won Do Heo[1,2,3]

Spatiotemporal control of gene expression or labeling is a valuable strategy for identifying functions of genes within complex neural circuits. Here, we develop a highly light-sensitive and efficient photoactivatable Flp recombinase (PA-Flp) that is suitable for genetic manipulation in vivo. The highly light-sensitive property of PA-Flp is ideal for activation in deep mouse brain regions by illumination with a noninvasive light-emitting diode. In addition, PA-Flp can be extended to the Cre-lox system through a viral vector as Flp-dependent Cre expression platform, thereby activating both Flp and Cre. Finally, we demonstrate that PA-Flp–dependent, Cre-mediated $Ca_v3.1$ silencing in the medial septum increases object-exploration behavior in mice. Thus, PA-Flp is a noninvasive, highly efficient, and easy-to-use optogenetic module that offers a side-effect-free and expandable genetic manipulation tool for neuroscience research.

[1] Department of Biological Sciences, Korea Advanced Institute of Science and Technology (KAIST), Daejeon 34141, Republic of Korea. [2] Center for Cognition and Sociality, Institute for Basic Science (IBS), Daejeon 34126, Republic of Korea. [3] KAIST Institute for the BioCentury, KAIST, Daejeon 34141, Republic of Korea. Correspondence and requests for materials should be addressed to B.O.P. (email: bopark@ibs.re.kr) or to W.D.H. (email: wondo@kaist.ac.kr)

Studies of complex brain functions require highly sophisticated and robust technologies that enable specific labeling and rapid genetic modification in live animals. A number of approaches for controlling the activity of genes or proteins in a spatiotemporal manner using light, small molecules, hormones and peptides have been developed for manipulating intact circuits or functions[1–5]. Cre/loxP recombination employing chemically inducible systems such as Cre-ER[T2], DD-Cre, and inducible promoter-directed Cre is the most commonly used in vivo gene-modification system[6–9]. Other approaches include selective or conditional Cre-activation systems within subsets of green fluorescent protein (GFP)-expressing cells (Cre-DOG)[10] or dual-promoter-driven intersectional populations of cells[11]. However, these methods are limited by the considerable time and effort required to establish knock-in mouse lines and by constraints on spatiotemporal control, which relies on a limited set of available genetic promoters and transgenic mouse resources.

Beyond these constraints, optogenetic approaches enable to control the activity of genetically defined neurons in the mouse brain with high spatiotemporal resolution. Recently, two optogenetic modules of photoactivatable-Cre recombinase (PA-Cre) have been developed using split-Cre components, one in which each component is fused to CRY2 and CIB1[12–14], and the other in which they are fused to positive Magnet (pMag) and negative Magnet (nMag)[15]. In both cases, illumination with blue light induces heterodimerization. To date, however, an optogenetic module for gene-manipulation capable of revealing spatiotemporal functions of specific target genes in the mouse brain has remained out of reach.

Cre/loxP recombination has been established as an efficient genetic manipulation system in mammals, and constitutes the overwhelmingly prominent technology underlying currently existing conditional gene knock-out/in mouse resources[16]. Applications of the Flp/Frt system to sophisticated genetic manipulations in vivo, including in brain tissue, have developed at an accelerating pace[17,18] since most recent improvement in Flp recombinase to produce a thermostable, codon-optimized version (*Flpo*) that functions efficiently in mammals[19–21]. The combinatorial use of Cre and Flp in Cre/Flp lines, in which expression is driven by tissue- or cell-type-specific promoters, has enabled more detailed investigations of different cell populations through conditional or selective genetic mutagenesis in vivo[16,22]. To date, however, no such light-inducible Flp system has been developed. Accordingly, we sought to develop a photoactivatable Flp recombinase (PA-Flp) that takes full advantage of the high spatiotemporal control offered by light stimulation.

Here, we demonstrate that the utility of PA-Flp as a non-invasive in vivo optogenetic manipulation tool for use in the mouse brain, even applicable to deep brain structures reaching hippocampus or medial septum (MS) by external LED light illumination (1–2 mW mm$^{-2}$). Furthermore, we engineer a Flp-dependent Cre driver as a module without leaky Cre expression in viral vector system, finally showing noninvasive light-dependent, Cre-mediated $Ca_v3.1$ gene silencing in MS neurons, which results in increased objective exploration behavior.

## Results

### Development of PA-Flp.
In designing PA-Flp, we employed a strategy in which Flpo was split into two pieces that reassemble in response to a light stimulus (Fig. 1a). Since a possible site at which Flpo could be split so as to restore activity upon reconstitution has not yet been reported, we first empirically addressed this question. To this end, we considered nine, aqueous-exposed loop regions of Flpo as possible sites, based on the structure of the full-length protein (Supplementary Figure 1a-c). Next, we sought

to introduce an efficient light-responsive module fused to each split-Flp site that satisfied the following four criteria: (1) light-inducible heterodimerization, (2) utilization of endogenous cofactors present at sufficient levels in mammals, (3) relatively small size to allow packaging in an adeno-associated virus (AAV), and (4) a high-affinity light-dependent interaction that minimally perturbs reassembly of split Flp. On the basis of these criteria, we selected the Magnet system (17.1 kDa)[23] as a suitable platform. This system offers a symmetry advantage compared with other light-responsive modules, such as CRY2 (65-70 kDa)-CIB1(21-22 kDa) pairs, which employ binding partners that are unbalanced in size.

In initial screens, we constructed fusions of each of the nine split-site variants of Flp fused at their C-terminus with nMagnetHigh1 (nMagH; nMagnet variants) or pMagnetHigh1 (pMagH; pMagnet variants), which are the strongest light-dependent dimerization pair among Magnet system[23]. To assess Flp activity in HEK293T cells, we used a Flp reporter plasmid in which FRT-stop-FRT-GFP is driven by a CMV promoter that produces a GFP signal upon catalytic activation of Flp. In these screens, we found that blue light-emitting diode (LED) illumination (470 ± 10 nm, 5-s pulse duration at 3-min intervals for 24 h) slightly increased Flp activity of split-site 1 (sp1:FlpN27/FlpC28) and split-site 7 (sp7:FlpN169/FlpC170) variants compared with the corresponding variants incubated in the dark (Supplementary Figure 1d). To further optimize the configuration, we next constructed sp1- and sp7-conjugated Magnet in both orientations, with or without a nuclear localization signal (NLS). Surprisingly, C-terminally fused split-FlpC showed dramatically increased (5- to 10-fold) recombination efficiency compared with N-terminally fused split-FlpC (see Supplementary Figure 1e compared with Supplementary Figure 1d). We termed sp1- and sp7-series with FlpNX-nMagH and pMagH-FlpCX orientation as PA-Flp1 or PA-Flp7, respectively. Additionally, we fused NLS N-terminally on PA-Flp1 or PA-Flp7 sites of FlpNX-nMagH, which showed that the potent NLS-tagging site for better efficacy is PA-Flp1 (Supplementary Figure 1d,e). Since, the efficiency of heterodimerization of between pMag and nMagH pair was comparable with pMagH and nMagH pair[23], we also tested pMag among PA-Flp1 variants. Finally, we found that among (NLS)-Nflp1-nMagH with (NLS)-pMag(H)-(NLS)-Cflp1-(NLS) tested, NLS-Nflp1-nMagH with pMagH-Cflp1 showed the highest light-dependent Flp activity (Supplementary Figure 1e,f). Notably, sp1 of split-Flp could also be used with other heterodimerization modules, such as FKBP-FRB or PHR-CIBN, suggesting the universality of this split site (Supplementary Figure 1g). To further characterize the efficiency of PA-Flp1 or 7 variants, we tested them in stable cell lines bearing Frt-STOP-Frt to concurrently validate genome accessibility and light sensitivity. Ultimately, we chose one of PA-Flp1 variants, NLS-Nflp1-nMagH with pMagH-Cflp1 pair (hereafter simply PA-Flp), which showed high light sensitivity (EC$_{50}$ = 3.1 μW cm$^{-2}$), fast recombination kinetics ($t_{1/2}$ = 1.1 h), and 2- to 8-fold induction compared with the other PA-Flp1 or PA-Flp7 variants tested (Supplementary Figure 2a, b).

### Performance of PA-Flp in vivo.
To verify PA-Flp in vivo, we electroporated PA-Flp with fDIO-YFP plasmids in embryonic day 15 (E15) embryonic mouse brains. Light illumination at P1 induced a substantial increase in Flp reporter (GFP) signal 2 days later compared with that observed in the dark (Fig. 1b). We also verified delivery of PA-Flp via an AAV vector in WT mice by co-infecting PA-Flp with fDIO-YFP, an exogenous Flp reporter, demonstrating a 30.4-fold increase in Flp reporter signal in the light-stimulated group compared with the non-light-stimulated

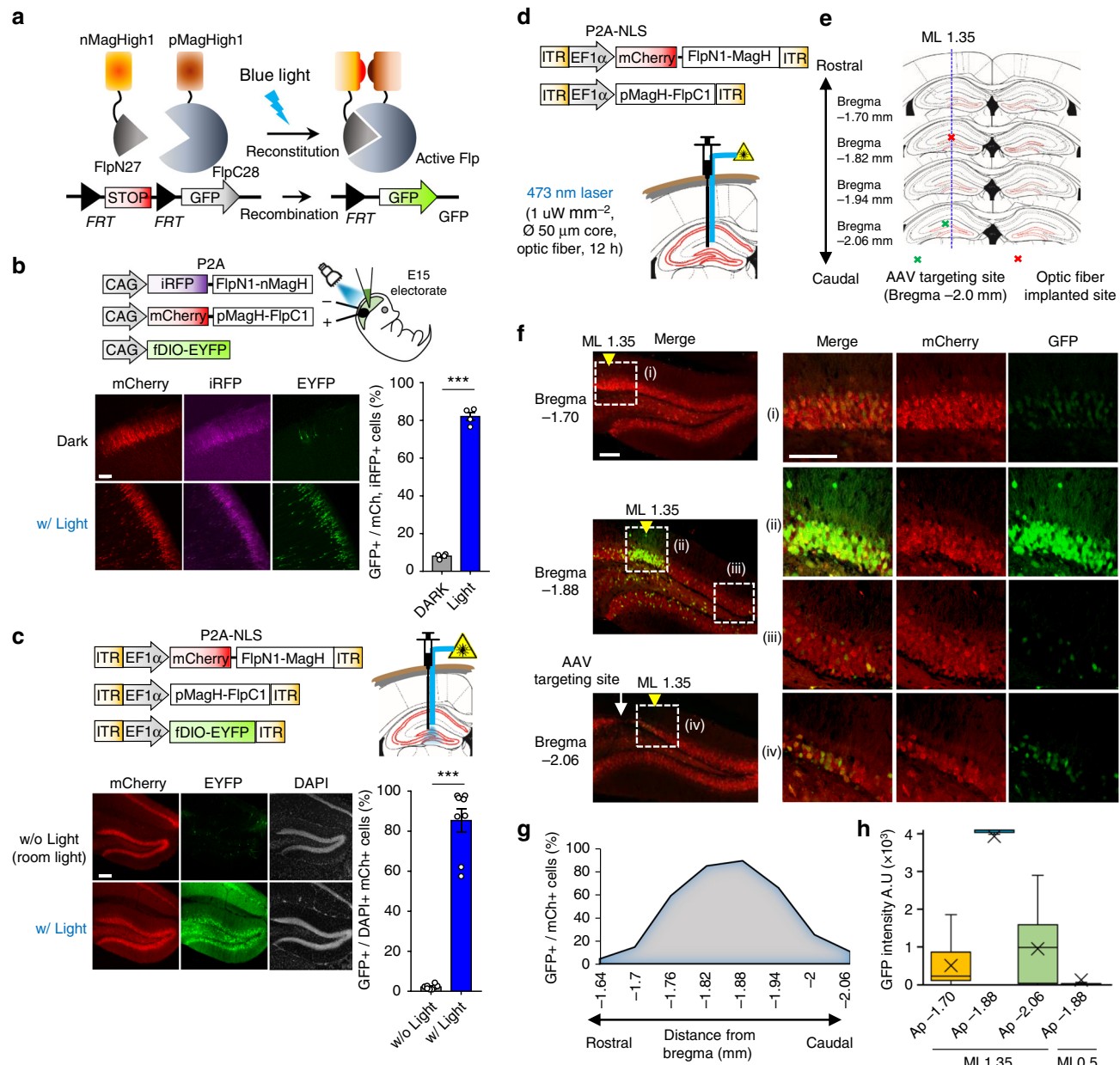

**Fig. 1** Development of PA-Flp. **a** Schematic depicting PA-Flp reconstitution and activation upon blue light illumination, and detection of GFP signals by PA-Flp-mediated deletion of a stop cassette in a Frt-floxed construct. **b** PA-Flp and fDIO-YFP (Flp reporter) expression plasmids were electroporated into an embryonic mouse brain (E15), with (w/) subsequent noninvasive light stimulation (0.5 mW mm$^{-2}$, blue fluorescent gun) at postnatal day 1–2 (P1–2) or maintained in dark conditions. Pup brains were harvested at P3-4. **c** AAV-EF1a-PA-Flp and AAV-EF1a-fDIO-YFP were co-infected into the hippocampus of 8-wk-old mice, with 30 min light (0.4 mW mm$^{-2}$, 20 Hz, 20% duty cycle) or without (w/o) light stimulation 2 wks after infection and sacrificed 1 wk after light stimulation. A blue line indicates laser light path through implanted optic fiber. **b, c** Measurement of GFP positive cells among both mCherry and iRFP positive (GFP+/mCh+iRFP+) cells or among both DAPI and mCherry positive (GFP+/DAPI+mCh+) cells in 4–8 coronal slices at each group. Scale bar: 100 μm. Data represent means ± s.e.m. ($n = 2$ mice/group; ***$P < 0.0001$, two-tailed Student's $t$-test). **d, e** Schematic depicting AAV infection in the mouse hippocampus and local light stimulation via implanted optic fiber (**d**), and AAV targeting sites (green) and optic fiber implantation sites (red) (**e**), in coronal sections of hippocampus. **f** Left: representative images showing local labeling of GFP signals from RCE:FRT mice at different depths along the rostral-caudal (RC) axis. Yellow arrowhead indicates the medial-lateral (ML) 1.35 coordinate. Right: sites (i)–(iv) at left marked in white-dashed squares are shown in higher-magnification views in the correspondingly labeled rows. Sites (i), (ii), and (iv) indicate the same ML coordinate along the RC axis; (iii) indicates a distal site with a different ML coordinate on the same coronal section of bregma (−1.88 mm). Scale bar: 100 μm. **g, h** Analysis of percentage (**g**) and intensity (**h**) of GFP signals along the ML axis at the same RC coordinate (bregma −1.88 mm) or RC axis on the same ML coordinate (ML 1.35). Measurement performed in each coronal slice. Data represent means ± s.e.m. ($n = 1$ mouse)

group (Fig. 1c). We then evaluated whether PA-Flp is capable of manipulating endogenous target sites with high efficiency using a Flp reporter mice line (RCE:FRT, Gt(ROSA)26Sor$^{tm1.2(CAG-EGFP)}$$^{Fsh}$/Mmjax)[24]. To assess the effects of light-delivery conditions on the performance of PA-Flp in RCE:FRT mice, we delivered blue light (400μWmm$^{-2}$) via an optic fiber for varying illumination times (5 s to 30 min). PA-Flp efficiency increased with increasing duration of light exposure up to 5 min, at which point efficiency saturated (Supplementary Figure 3b,c).

**'Local genetic labeling' in the mouse brain by precise light targeting**. Since the highly light-scattering nature of brain tissue requires more precise light-targeting conditions[25], we sought to minimize light diffusion/scattering using thin optic fibers (50–60 μm) with varying laser intensity, frequency (Hz), and duty cycle (%) (Supplementary Figures 3d,e and 4; see Methods for details). Applying these parameters in an optimization strategy, we validated the concept of 'local genetic labeling' in the hippocampal dentate gyrus (DG) and the M1 cortex (Supplementary Figures 5 and 6). We found that it was possible to adjust labeling over a relatively broad range (300–400 μm) with gradients (Supplementary Figure 5a, b) and to a smaller range (~100 μm) with sharper patterns (Supplementary Figure 5c) in hippocampal DG. To verify three-dimensional labeling patterns, we analyzed the dorsal DG along the rostral-caudal and medial-lateral axes in every coronal section in the dorsal DG. Through demonstrated that three-dimensionally restricted (~200 μm), anatomically defined areas could be genetically manipulated (Fig. 1d–h).

**PA-Flp activation by transcranial LED illumination**. To maximize the utility of the highly light-sensitive characteristics of PA-Flp, we tested its action by noninvasive light without prior implantation of an optic fiber. We found that light illumination using a light guide coupled LED (Ø 0.6 cm) at an intensity of 50 μW mm$^{-2}$ for 30 min effectively delivered light to the cortex regions of mouse brain and produced highly efficient recombination performance (Supplementary Figure 7). Considering these results together with the captured images of noninvasive LED light penetrating through the mouse brain (Supplementary Figure 8), we anticipated that stronger light power (up to 5 mW mm$^{-2}$) would provide sufficient light to activate PA-Flp at the depth of the hippocampus. Following PA-Flp expression in the hippocampus, we illuminated the mouse head with LED light at intensities of 0.1, 1, and 5 mW mm$^{-2}$ for 30 s or 30 min with the fur removed, leaving the skull and skin intact. Surprisingly, we found that LED illumination with an intensity as low as 1 mW mm$^{-2}$ for 30 s was sufficient for full activation of PA-Flp in the hippocampus (Fig. 2a–c). Stronger LED intensity (~5 mW mm$^{-2}$) or increased illumination time (~30 min) did not lead to proportionally higher PA-Flp efficiency, indicating that LED illumination at 1 mW mm$^{-2}$ for 30 s provided sufficient light to induce maximal PA-Flp activity (Supplementary Figure 9). To test PA-Flp activation in regions of the brain deeper than the hippocampus, we illuminated the medial septum (MS), a region 3.5–4.0 mm distal from the skull surface. These experiments showed that LED illumination at 2 mW mm$^{-2}$ for 30 s was sufficient to activate PA-Flp in the MS (Fig. 2d–f).

**Extended utility of PA-Flp to Cre-loxP system**. To extend the applicability of PA-Flp to the many existing loxP-flanked (floxed) lines, we investigated how to activate recombination at loxP sites using PA-Flp. To this end, we designed a Flp-dependent Cre driver (FdCd) by inserting Cre into an fDIO cassette (fDIO-Cre). In initial experiments employing transient plasmid expression in vivo, we found that Flp activity was effectively transferred to

activation of Cre (Supplementary Figure 10). However, packaging this fDIO-Cre construct in AAV or lentivirus resulted in unexpectedly high basal Cre activity in both cultured neurons and mouse brains (Supplementary Figure 11b-d). We hypothesized the viral nascent ITR/LTR promoter-driven transcripts with antisense direction generated a substantial number of Cre molecules to react with substrate DNA sequences under long-expression conditions, despite weakness of the ITR/LTR promoters[26–28]. To solve this problem, we designed a new FdCd candidate to block translation of ITR/LTR promoter-driven Cre (Supplementary Figure 11a). A strategy repositioning the NLS and Kozak sequence outside of the upstream of fDIO cassette dramatically decreased the expression of Cre derived from antisense transcripts (Supplementary Figure 11a-e). Through this strategy, antisense transcripts lost both strong kozak and functional NLS sequences of Cre in basal state, but sense transcripts (target promoter-driven NLS-tagged Cre)-mediated expression is fully induced after Flp activation. Additionally, we introduced point mutations (Methionine to Leucine) on putative Kozak consensus sequences[29,30] at amino acid position 11th and 13th of Cre, which has no impact on its catalytic activity. The final version which repositioned the NLS and Kozak sequences with the dual mutations of Cre showed the least basal Cre activity among tested candidates in cultured neurons (Supplementary Figure 11b, c) or in vivo brain (Supplementary Figure 11d,e), while robustly increasing Cre activity following Flp activation (Fig. 3a–f and Supplementary Figures 12 and 13). We termed this candidate, Leak-Free FdCd (LF-FdCd).

**Applications of PA-Flp-dependent Cre system**. To evaluate applications of the PA-Flp–dependent Cre (PA-FdCre) system to neurobehavioral research, we targeted mixtures of AAVs expressing PA-Flp, LF-FdCd, and floxed-stopped–shCa$_v$3.1 (LoxP-STOP-LoxP flanked small hairpin RNA [shRNA] targeting Ca$_v$3.1) to the MS of wild-type mice (Fig. 4a). Our previous study revealed that silencing Ca$_v$3.1 T-type calcium channels in GABAergic neurons in the MS that project to the hippocampus increased object-exploration behavior in mice[31]. We thus validated this previous finding by silencing Ca$_v$3.1 specifically in the MS using light-triggered PA-FdCre. To evaluate object-exploration behavior, we first injected mice with PA-FdCre and floxed-stopped–shCa$_v$3.1, and then exposed them to LED illumination (shCa$_v$3.1 w/ LED mice) or room light conditions (shCa$_v$3.1 w/o LED mice); as a control, mice were injected with PA-FdCre and floxed-stopped–shControl (scrambled control shRNA construct) followed by LED illumination (shControl w/ LED mice) or room light conditions (shControl w/o LED mice). Thereafter, mice were exposed to novel objects in a familiar arena and monitored for 20 min (Fig. 4b). Tracking of mouse movements revealed enhanced object-exploration behavior by ShCa$_v$3.1 w/ LED mice compared with shCa$_v$3.1 w/o LED, shControl w/ LED, and shControl w/o LED mice (Fig. 4d), as confirmed by measurements of total exploration (Fig. 4e, f). Consistent with our previous observation[31], the locomotor activity of shCa$_v$3.1 w/ LED mice, measured as total distance moved, was not significantly different from that of shCa$_v$3.1 w/o LED or shControl w/ LED, and shControl w/o LED mice during the habituation period of the object-exploration task (Fig. 4g). To assess the extent of gene silencing, we sacrificed mice after the completion of behavioral tests and measured Ca$_v$3.1 expression in MS neurons using immunohistochemistry. The proportion of Ca$_v$3.1-positive neurons in the MS decreased in shCa$_v$3.1 w/ LED mice compared with shCa$_v$3.1 w/o LED (Fig. 4c), indicating that PA-FdCre–mediated shCa$_v$3.1 silencing reduced endogenous Ca$_v$3.1 expression in the MS.

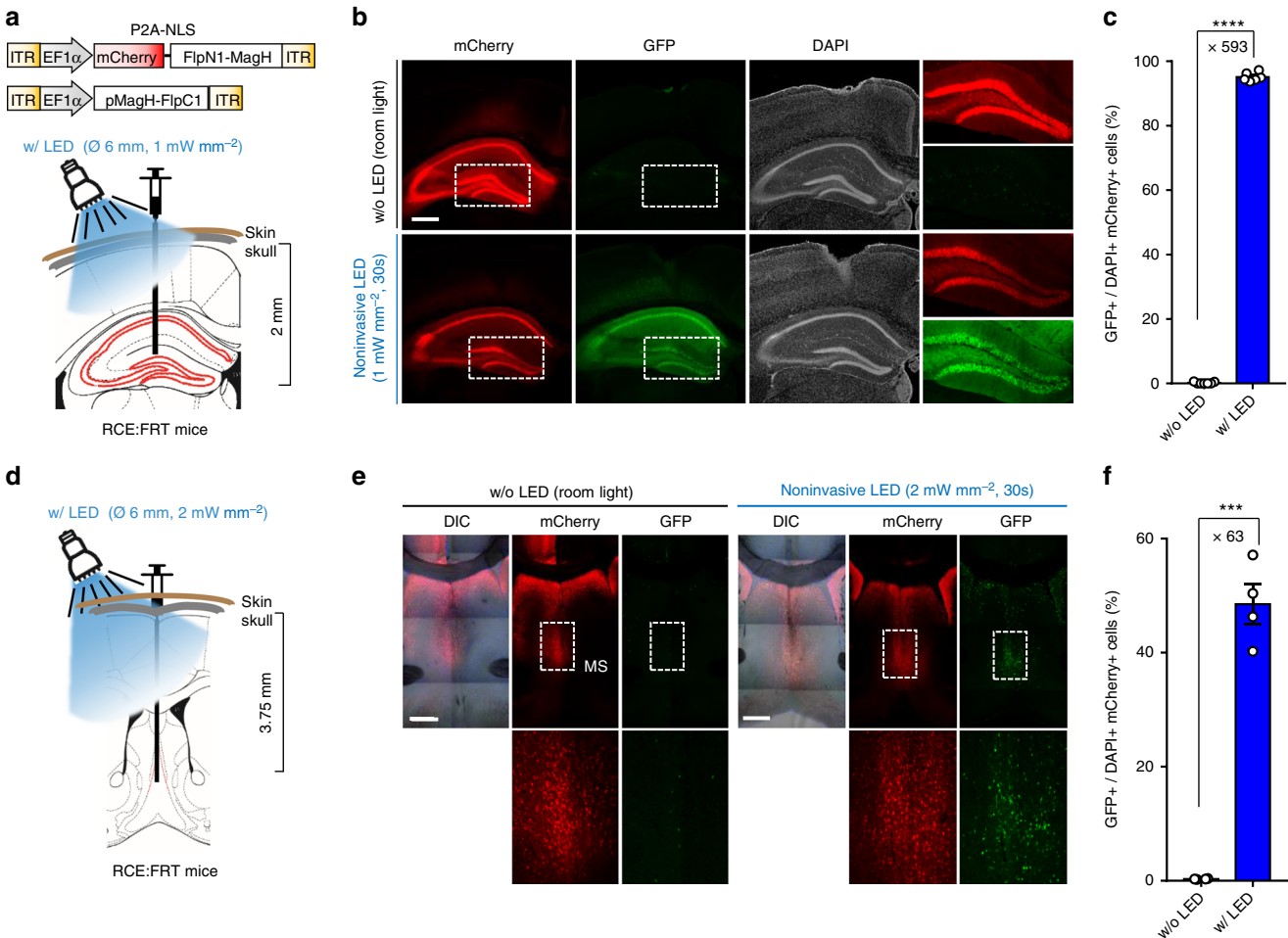

**Fig. 2** Noninvasive LED illumination activates PA-Flp in deep brain structures down to the hippocampus and MS. **a**, **d** Schematic depicting AAV-EF1a-PA-Flp targeting in the hippocampus (**a**) or MS (**d**) followed by LED illumination. **b**, **e** Representative images of mCherry (PA-Flp) and GFP (Flp reporter) signals from RCE:FRT mice (8–12-wk-old), with (w/) and without (w/o) LED illumination. Two wks after infection, light was illuminated noninvasively with white LED at an intensity of 1 mW mm$^{-2}$ (**b**) or 2 mW mm$^{-2}$ (**e**) for 30 s through the intact skull and skin. Mice were maintained under room light as described in Methods. All mice were sacrificed 3 wks after infection. Scale bar: 500 μm. **c**, **f** GFP positive cells among both DAPI and mCherry positive (GFP+/DAPI+mCh+) cells were measured in the hippocampus (**c**) and MS (**f**) region of 4–7 coronal slices, as shown in **b** and **e**, respectively. Data represent means ± s.e.m. (**c**, $n = 4$ mice/group, ****$P < 1 \times 10^{-10}$; **f**, $n = 2$ mice/group, ***$P < 0.0001$; two-tailed Student's $t$-test)

## Discussion

In this study, we developed PA-Flp by searching out new split sites of Flp recombinase that was not previously identified, being capable of reconstitution to be active. We validated the highly light-sensitive, efficient performance of PA-Flp through precise light targeting by showing transgene expression within anatomically confined mouse brain regions. The concept of 'local genetic labeling' presented here suggests a new approach for genetically identifying subpopulations of cells defined by the spatial and temporal characteristics of light delivery. Importantly, genetic labeling is a permanent marking system that enables tracking of locally labeled subpopulations of cells, such as migrating, differentiating or proliferating cells, without loss of the labeled signal.

PA-Flp activation through noninvasive light illumination deep inside the brain is advantageous in that it avoids chemical- or optic fiber implantation-mediated side effects, such as off-target cytotoxicity or physical lesions, that might influence animal physiology or behaviors[32,33]. Our optogenetic system is very easy to set up and simple to use, requiring only an LED (blue or cool-white) source that satisfies light power and amount criteria, indicated above, for application to deep mouse brain regions, and it does not necessitate injury to the skull or skin for light delivery.

Finally, our technique provides expandable utilities for transgene expression system stringently upon Flp activity in vivo, by designing a viral vector for minimal 'leaky' expression influenced by viral nascent promoters, such as ITRs or LTRs, located at 5′ or 3′ end in AAV or Lentivirus. Accordingly, we showed that the PA-Flp-dependent Cre expression system called PA-FdCre is an expandable modality of PA-Flp, demonstrating the applicability of a light-inducible, Cre-dependent RNAi system to neurobehavioral research. The ability to support applications involving loxP-containing mouse lines or vector systems further highlights the versatility of PA-FdCre. Furthermore, PA-FdCre is a module of light-inducible Flp- and Cre-dual activation system, but Cre expression is activated by Flp activity along with the neuron-specific promoter-directed LF-FdCd as an intersectional manner. According to our observations, the probability of leaky Flp activity (attributed by spontaneous auto-assembly of split-Flp components) is far less likely to occur in driving two orthogonal reactions of Flp-FRT. Therefore, the design of gene expression cassette turned ON only under the sequential or intersectional Flp and Cre activation events[21] would offer greater reliability with robust amplitude of inducible gene regulation systems in vivo. Taken together, our findings indicate that PA-Flp is a highly

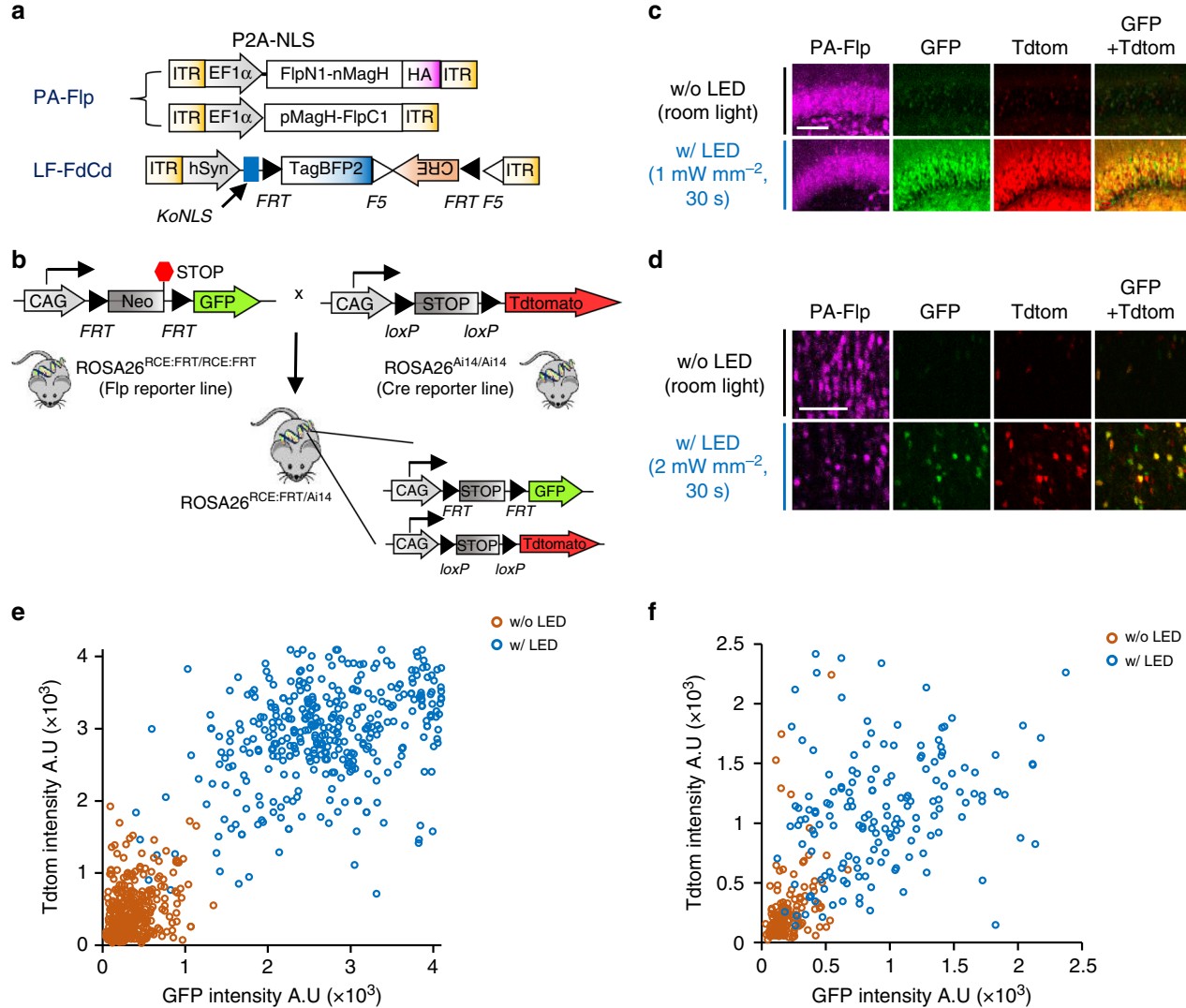

**Fig. 3** Verification of PA-Flp-dependent Cre (PA-FdCre) system in ROSA26[RCE:FRT/Ai14] mouse line. **a** Schematic depicting AAVs carrying PA-Flp and Leak Free Flp-dependent Cre driver (LF-FdCd) as PA-Flp dependent Cre (PA-FdCre) systems. **b** Schematic depicting the generation of ROSA26[RCE:FRT/Ai14] mice. Flp reporter (GFP) and Cre reporter (TdTomato) alleles are located on the ROSA26 cassette in ROSA26[RCE:FRT/Ai14] mice. **c, d** Light-induced dual Flp and Cre activation in the hippocampal DG (**c**) or MS (**d**) of ROSA26[RCE:FRT/Ai14] mice (8–12-wk-old). Scale bar: 100 μm. **e, f** Correlation between GFP and TdTomato fluorescence intensity in the hippocampal DG (**e**) or MS (**f**) among PA-Flp–positive cells (magenta) in with (w/) LED and without (w/o) LED groups. A total of 200–500 cells were counted from each w/ LED and w/o LED group (n = 4 mice/group)

efficient genetic manipulation tool that is capable of providing a noninvasive optogenetic module with greatly expandable applications in neuroscience research.

## Methods

**Design and construction of PA-Flp**. DNA and amino acids sequences of notable constructs used in this study are described in detail in Supplementary Note 1. Most constructs were generated with Gibson Assembly Cloning (New England Biolabs) using previously generated molecular biology reagents as PCR templates. The platform for screening PA-Flp series was created by first generating a pQC-mCherry-IX vector using standard cloning techniques. An *Age*I/*Bgl*II-digested fragment containing mCherry from the pmCherry-C1 vector (Clontech) was inserted into the pQCXIX vector between *Age*I and *Bam*HI sites, placing it under control of a CMV promoter. Each pair of PA-Flp recombinase series was generated by Gibson assembly Cloning. Split Flp fragments were amplified from the coding sequence of improved Flp (FLPo), obtained from Addgene plasmids (ID:55634). nMagHigh and pMagHigh were synthesized by Genescript. Each PCR fragment was inserted into a pQC-mCherry-IX vector at a single *Eco*RV restriction site under the control of an IRES in different configurations. The sequence encoding Nflp1 and nMagH was PCR-amplified using each primer pair 5′-CAC GCG TCT CGA GAT ATC ACC ATG GCT CCT AAG AAG AAG AGG-3′ (forward) and 5′-GCC ACC TCC GCC TGA ACC GCC TCC ACC GCT GGG CCT CTC GAA T-3′

(reverse) or 5′-AGG CGG AGG TGG CAG CGG CGG TGG CGG ATC GCA CAC CCT GTA CGC CCC CG-3′ (forward) and 5′-GCC TGG ACC ACT GAT ATC TTA CTC GGT CTC GCA CTG GAA GC-3′ (reverse), respectively. The PCR fragments are reacted with *Eco*RV-linearized pQC-mCherry-IX vector by Gibson assembly to generate pQC-mCherry-IRES-Nflp1-nMagH. The sequence encoding Cflp1 and pMagH was PCR-amplified using each primer pair 5′-CAC GCG TCT CGA GAT ATC ACC ATG CAC ACC CTG TAC G-3′ (forward) and 5′-GAA CCG CCT CCA CCC TCG GTC TCG CAC TGG-3′ (reverse) or 5′-CAC GCG TCT CGA GAT ATC ACC ATG CCC AAG AAG AAG AGG AAG GT GGG CGA GAA GAT CGC CAG CGG ATC CAC CTC CGC CTG TGA TGT AGC AGT GC-3′ (forward) and 5′-GCC TGG ACC ACT GAT ATC TTA GAT CCG CCT GTT GAT GTA GCT G-3′ (reverse), respectively. The PCR fragments are reacted with *Eco*RV-linearized pQC-mCherry-IX vector by Gibson assembly to generate pQC-mCherry-IRES-pMagH-Cflp1. Gibson homology region, glycine-serine linker, HA tag, P2A, and NLS sequences, obtained from SV40 sequences, were introduced using PCR primers. AAV-EF1a-mCherry-FlpN1-nMagH and AAV-EF1a-pMagH-FlpC1 were generated by PCR-amplification of each fragment from pQC-mCherry-IRES-Nflp1-nMagH and pQC-mCherry-IRES-Cflp1, respectively, followed by cloning into *Bsp*EI and *Eco*RI sites of the pAAV-Ef1a-DIO EYFP vector by Gibson assembly. For in utero electroporation, pCAG-iRFP-P2A-FlpN1-nMagH and pCAG-mCherry-P2A-pMagH-FlpC1 constructs were prepared by assembling PCR fragments from mCherry, iRFP, PA-FlpN and PA-FlpC at the *Bsp*EI/*Eco*RI site of the pCAG-fDIO-EYFP vector. pCAG-fDIO-EYFP and pCAG-DIO-EGFP

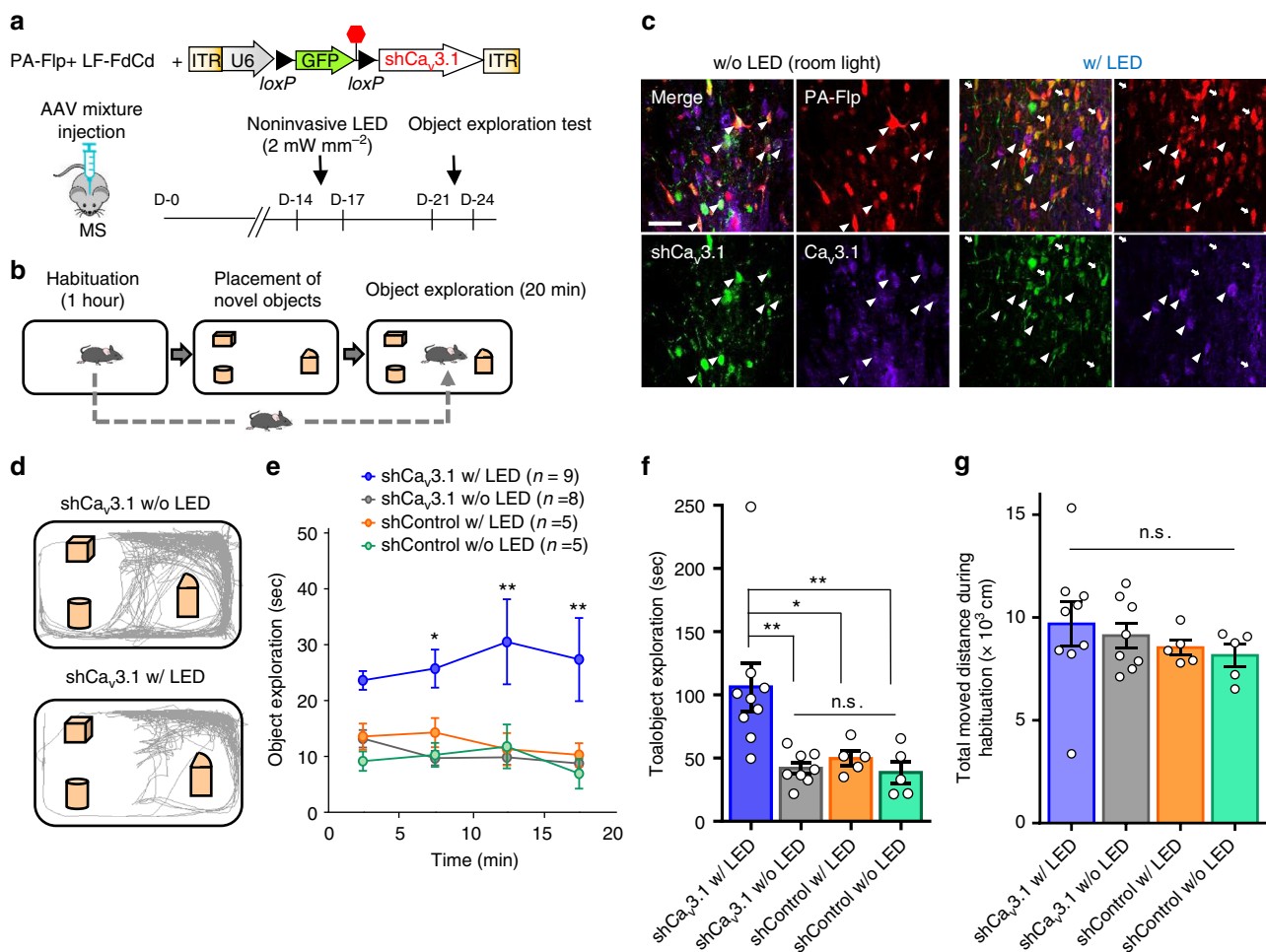

**Fig. 4** PA-FdCre–mediated Ca$_v$3.1 knockdown in the MS increases object exploration. **a** Injection of a mixture of AAVs expressing PA-Flp, LF-FdCd and flox-shCa$_v$3.1 (or flox-shControl) into the MS of a 8-wk-old mice brain, with (w/) or without (w/o) subsequent noninvasive white LED (2 mW mm$^{-2}$) illumination 2 wk after infection. **b** Schematic depicting object-exploration behavior in PA-FdCre–mediated, MS-specific, Ca$_v$3.1-knockdown mice. **c** Immunohistochemical detection of Ca$_v$3.1 in the MS following co-infection with PA-Flp-, LF-FdCd- and flox-shCa$_v$3.1-expressing AAVs. PA-Flp (mCh) and flox-shCa$_v$3.1 (GFP) fluorescence, Ca$_v$3.1 immunofluorescence (violet), and merged images are presented. Arrowheads indicate Ca$_v$3.1-positive cells. Arrows indicate PA-Flp/flox-shCa$_v$3.1–positive, Ca$_v$3.1-negative cells. Scale bar: 50 μm. **d** Cumulative traces of representative navigation pathways of shCa$_v$3.1 with LED and shCa$_v$3.1 without LED mice during object-exploration behavior. **e** Increased object-exploration time for shCa$_v$3.1 w/ LED mice ($n = 9$/group) compared with shCa$_v$3.1 w/o LED mice ($n = 8$; $P = 0.003$, two-way repeated measures ANOVA; $P = 0.004$, Bonferroni post-hoc test), shControl w/ LED mice ($n = 5$; $P = 0.003$, two-way repeated measures ANOVA; $P = 0.028$, Bonferroni post-hoc test), and shControl w/o LED ($n = 5$, $P = 0.003$, two-way repeated measures ANOVA; $P = 0.009$, Bonferroni post-hoc test). **f** Increased total object-exploration time for shCa$_v$3.1 w/ LED mice ($n = 9$) compared with that for shCa$_v$3.1 w/o LED mice ($n = 8$; $P = 0.003$, one-way ANOVA; $P = 0.004$, Bonferroni post-hoc test), shControl w/ LED mice ($n = 5$; $P = 0.003$, one-way ANOVA; $P = 0.029$, Bonferroni post-hoc test), and shControl w/o LED mice ($n = 5$; $P = 0.003$, one-way ANOVA; $P = 0.008$, Bonferroni post-hoc test). **g** Locomotor activity of shCa$_v$3.1 w/ LED ($n = 9$), shCa$_v$3.1 w/o LED ($n = 8$), shControl w/ LED mice ($n = 5$), and shControl w/o LED ($n = 5$) mice in the experimental cage during the 60-min habituation period of the object-exploration task ($P = 0.612$, one-way ANOVA). **e**–**g** Data represent means ± s.e.m. (*$P < 0.05$, **$P < 0.01$)

were cloned into the *Eco*RI/*Not*I site of the pCAG-C1 vector using PCR-amplified fragments from pAAV-EF1a-fDIO-EYFP (Addgene ID:55641) and pAAV-EF1a-DIO-EGFP (Addgene ID:37084), respectively. LF-FdCd was generated by introducing Kozak and NLS sequences in front of the fDIO-Cre cassette by PCR amplification. The hSyn promoter was PCR-amplified from the pAAV-hSyn-DIO-EGFP vector (Addgene ID:50457).

**Stable cell line**. A HEK293T cell line stably expressing Frt-stop-Frt-EGFP was produced by transduction of a lentiviral construct, prepared by PCR-amplifying a fragment from pCAFNF-GFP (Addgene ID:13772) and cloning it into a pLenti vector under the control of an EF1a promoter by Gibson assembly. Lentivirus was packaged by co-transfection of HEK293T cells on 15-cm plates with VSVG, pDelta, and payload constructs using polyethylenimine (#23966-2, Polyscience). After 4 h, the medium was changed and cells were cultured for an additional 72 h. Culture supernatants were collected at 48 and 72 h, pooled, and filtered through a 0.45-μm filter. HEK293T cells were transduced with lentivirus particles, and stable clones

were selected by culturing with 300 μg/ml G418 (Geneticin) beginning 48 h after transduction. Single-cell clones of transduced HEK293T stably expressing Frt-stop-Frt-EGFP were collected by pipetting, after which recombination of Frt-stop-Frt-EGFP in each single cell clone lines was verified by transfection of Flp recombinase.

**Cell culture, transfection, and live-cell imaging**. HEK293T cells were acquired from the American Type Culture Collection (ATCC, Manassas, VA, USA) and were assessed for contamination using a PCR-based mycoplasma detection kit (Cell-Safe, BioMycoX). HEK293T cells were grown in high-glucose Dulbecco's Modified Eagle Medium (DMEM; PAA Laboratories GmbH) supplemented with 10% FBS (Invitrogen) and 1% penicillin-streptomycin (Invitrogen). Cells were incubated at 37 °C in a humidified 10% CO$_2$ incubator. For experiments, dissociated HEK293T cells were plated at a density of $1 \times 10^4$ cells/well in 96-well plates and transfected in duplicate with 200 ng total DNA using a Fugen6 transfection kit (Promega) according to the manufacturer's protocol. Cells were stained with 500 ng/ml Hoechst 33342 (Invitrogen) for 30 min at 37 °C before acquiring

images. Images were acquired at the indicated time points with an ImageXpress Micro XLS automated epifluorescence microscope (Molecular Devices) using a ×20 Plan Fluor objective and a 4.66 megapixel CMOS camera with a 16-bit readout. Image analysis was performed using MetaXpress software (Molecular Devices).

**In utero electroporation and light stimulation**. In utero electroporation was performed following the procedure as described in the literature[34]. Briefly, wild-type, timed-pregnant (embryonic day 15 [E15]) C57BL/6 mice were anaesthetized with isoflurane in an oxygen carrier. An endotoxin-free, purified 1–2 μl DNA solution (1 μg/μl) was injected into the brain ventricle of embryos using a pulled micropipette and Picospritzer III (#052-0500-900, Parker). DNA-injected embryos, held through the uterus using forceps-type electrodes (#45-0489, Tweezertrodes Kits), were delivered four to five electric pulses (duration, 50 ms), via an ECM 830 electroporator (#45-0052, BTX). On postnatal day 1–2, pups were remotely stimulated with light (1 mW mm$^{-2}$) three times (5 s each) over the course of 1 h using a DFP-1 Dual Fluorescent Protein Flashlight (440-460 nm; Nightsea) or were exposed to 30-s pulses of computer software-controlled blue LED illumination (470 nm peak, 250 μW mm$^{-2}$ [TouchBright; Live Cell Instrument]) at 3-min intervals for 2 h. Mice were sacrificed 2 d after light stimulation (postnatal days 3-4).

**AAV production**. AAVs (serotype DJ/8) were produced using a previously described three-plasmid cotransfection system[35]. Briefly, endotoxin-free solutions of purified helper plasmid (pHelper), packaging plasmid (pRC-DJ/8), and transfer plasmid (containing the transgene expression cassette) DNA were mixed at a 2:1:1 (μg) ratio (final concentration, 1–1.5 μg/μl). HEK293T cells were plated on a 15-cm dish (8 × 10$^6$ cells/dish) in 20 ml DMEM, and 24 h later were transfected with 0.75 μg/ml of a 2.5:1 mixture of DNA (μg)/PEI (μg) ratio in media using poly-ethylenimine. Seventy-two hours after transfection, cells were collected and resuspended in 14 ml lysis buffer (50 mM Tris–Cl pH 8.0, 150 mM NaCl, 2 mM MgCl$_2$), followed by addition of 10% sodium deoxycholate (final concentration, 0.5%) and benzonase (final concentration, 50 U/ml). Cell lysis solutions were incubated at 37 °C for 30 min, after which lysates were subjected to three freeze-thaw cycles and centrifuged. The resulting supernatant was loaded onto an iodixanol gradient and centrifuged at 69,000 rpm (350,000 × g) for 1 h at 4 °C. The 40% iodixanol fraction was isolated, washed with phosphate-buffered saline (PBS) in a 100,000 WMCO tube filter, and concentrated to 100–200 μl. AAV titers were measured using an AAVpro Titration Kit for Real Time PCR (#6233, TaKaRa); titers of all AAVs used for in vivo infection were in the range of 1 × 10$^{12}$ to 1 × 10$^{13}$ viral genomes (vg)/ml.

**Stereotaxic surgery and viral injection**. Mice were anesthetized with Avertin (240 mg/kg) or a mixture of ketamine (120 mg/kg) and xylazine (10 mg/kg). A cocktail of AAV-EF1a-(mCh-P2A)-NLS-FlpN-nMagH-(HA) and AAV-EF1a-pMagH-FlpC (1:1 ratio, 2.5 × 10$^8$ vg each), collectively termed AAV-EF1a-PA-Flp, was stereotactically co-injected with the Flp reporter, AAV-EF1a-fDIO-YFP (1.25 × 10$^8$ vg), into the hippocampal DG or CA1 regions of 8–12-wk-old C57BL/6 WT mice.

For Tg (RCE:FRT) experiments, the hippocampal DG, CA1 or MS regions of 8–12-wk-old mice were infected with AAV-EF1a-PA-Flp (5 × 10$^8$ vg) in a total volume of 1.0 μl; the M1 cortex was infected with a cocktail of AAV-CAG-mCh-P2A-Nflp1-nMagH and AAV-CAG-pMagH-CFlp1 (1:1 ratio, 1.25 × 10$^8$ vg each), collectively termed PA-Flp$_{\triangle NLS}$, in a total volume of 1.0 μl.

For Tg (RCE:FRT/Ai14) experiments, the hippocampal DG, CA1 or MS region was co-infected with AAV-EF1a-PA-Flp (5 × 10$^8$ vg, HA-tagged version) and AAV-hSynI (CamKIIa)-KoNLS-fDIO-Cre (MtoL) (LF-FdCd, 5 × 10$^7$ vg) in a total volume of 1.0 μl. All viruses were infused using a WPI 33 g blunt NonoFil needle at an infusion rate of 0.1 μl/min. To permit diffusion of the AAV mixture into brain tissue and prevent leakage through the needle tract, we held the needle in place for 10 min after completion of each injection. The stereotaxic coordinates were as follows: DG region: AP −2.0, ML 1.4, DV 1.7; CA1 region: AP −2.0, ML 1.4, DV 1.25; M1 region: AP 1.5, ML 1.5, DV 0.7; and MS region: AP 0.86, ML 0.0, DV 3.7.

AAVs carrying an shRNA outside two loxP sites (AAV-U6-LoxP-CMV-GFP-LoxP-shRNA, 1 × 10$^9$ vg) (KIST virus facility, KOREA) were used for Cre-dependent silencing of Ca$_v$3.1 in the MS. The shRNA oligonucleotides for targeting Ca$_v$3.1 mRNA (shCa$_v$3.1; 5′-CGG GAA CGG GAA GAT CGT AGA TAG CAA A-3′) and control shRNA (shControl; 5′-AAT CGC ATA GCG TAT GCC GTT-3′) were created following the information described in the literature[31,36]. Control shRNA sequences were used to construct a non-targeting control virus. Mice were given injections of either shCa$_v$3.1 or non-targeting shControl.

**Histology**. For fluorescence imaging, isolated mouse brains were fixed by incubating in 4% paraformaldehyde (PFA) in PBS for 1 d at 4 °C. Brains in ice-cold PBS were sectioned into 50–60 μm coronal slices using a VT1200S vibratome (Leica). Slices were mounted in Fluoromount G (Southern Biotech), with or without DAPI (4′,6-diamidino-2-phenylindole). For immunohistochemistry, mouse brains were transcardially perfused with PBS followed by fixation in 4% PFA overnight at 4 °C. Slices were incubated in blocking solution (5% normal goat serum, 0.3% triton-X in PBS) for 1.5 h and stained with primary antibody overnight at 4 °C. Anti-HA-Tag rabbit mAb (1:1000, #3724, CST) and anti-Ca$_v$3.1 (1:200, #ACC-021, Alomone

Lab) were used to detect HA-tagged PA-Flp and endogenous Ca$_v$3.1, respectively. After incubating with primary antibodies, slices were washed five times with 0.3% Triton-X in PBS, incubated with secondary antibody in blocking solution for 1.5 h at room temperature, and then washed five times in 0.3% Triton-X in PBS. Images were captured with a Nikon A1 confocal microscope.

**Animals**. C57BL/6J inbred mice were obtained from the Jackson Laboratory (JAX Mice and Services, Bar Harbor, ME, USA). The RCE:FRT Flp reporter line (Gt (ROSA)26Sor$^{tm1.2(CAG-EGFP)Fsh/Mmjax}$)[24], was purchased from the Jackson Laboratory as cryopreserved sperm. Heterozygous mice were obtained by in vitro fertilization, performed by the KAIST animal facility. The Ai14 Cre reporter line (Gt(ROSA)26Sor$^{tm14(CAG-tdTomato)Hze/J}$)[37] was a gift from Dr. Shin (Institute of Basic Science, Korea). ROSA26$^{RCE:FRT/Ai14}$ mice were generated by crossing homozygous RCE:FRT and Ai14 mice; these Flp and Cre reporter lines have been previously characterized[16,37–39]. Mice were maintained with free access to food and water under a 12/12-h light/dark cycle (light intensity in cages measured under 'room light' conditions, 2.5–5 μW mm$^{-2}$). Animal care and experimental procedures followed the guidelines of the Institutional Animal Care and Use Committee of Institute for Basic Science.

**Light source**. A Ø 200 μm or 50–60 μm optic fiber (Doric) coupled to a blue diode 473-nm laser (MBL-III-473m; CNI) was used to deliver blue light to broad or local regions, respectively, within the hippocampal DG (see Supplementary Figure 4 for details). Immediately after AAV injection, optic fiber ferrules were implanted in approximately the upper 200-μm region of AAV-targeted sites. Custom-made (LCI, Korea) external LEDs (white or 470 nm blue), excited through a 0.28 cm$^2$ coupled fiber and controllable up to a power intensity of 10 mW mm$^{-2}$, were used for noninvasive light stimulation (See Supplementary Figure 9b for details of LED performance in the mouse brain). Light intensity was measured at the surface of the optic fiber or LED coupled-fiber end using the photodetector of a power meter (#PM120D; Thorlab). The light intensity per unit area (mW mm$^{-2}$) from an optic fiber tip was defined as a measured value using the power meter.

**Optical settings for local light delivery**. Light stimulation of broad or local regions was performed using protocol settings described in Supplementary Figure 4a. Light-scattering effects of brain tissue were assessed by implanting optic fibers (Ø 200 μm) into the hippocampus of an anesthetized mouse and monitoring light-delivery patterns under various light-stimulation conditions using a dissecting microscope (Supplementary Figure 4b). The observation matched well with GFP (Flp reporter) signal patterns obtained from each light stimulation protocols. In addition to this, light diffusion emanating from the optic fiber tip was greatly reduced by introducing a tiny optic fiber (Ø 50–60 μm), combined with the use of optical settings with a low numerical aperture (NA; 0.1–0.22). Since total internal reflection occurs in conditions above critical angle within optic fibers, a lower NA results in the formation of more straightly oriented light path shapes (Supplementary Figure 4c), producing a light ray that exits the optic fiber tip with a lower diffusion angle (Supplementary Figure 4d). Tests of PA-Flp in the M1 region revealed a substantial amount of Flp leakage (~20%) under normal 12 h-light on/ off housing conditions, an effect that was likely attributable to the ultra light-sensitive characteristics of PA-Flp or possibly relatively high basal activity in the motor cortex cells where PA-Flp was expressed. Accordingly, we tested a PA-Flp version without the NLS tag (PA-Flp$_{\triangle NLS}$), which is less light sensitive than PA-Flp (Supplementary Figures 1e and 2a,b). Applied to the M1 region as a CAG promoter-driven form, PA-Flp$_{\triangle NLS}$ exhibited significantly decreased basal Flp activity in the cortex (~5%) compared with that of PA-Flp (~20%). Local genetic labeling in the M1 region was validated using PA-Flp$_{\triangle NLS}$. In these experiments, an optic fiber (Ø 60 μm) was implanted by placing it in the upper part of layer V (Supplementary Figure 6a); PA-Flp$_{\triangle NLS}$ expression was relatively broad across layers I–VI. Two light-intensity conditions (5 or 25 μW mm$^{-2}$) and two illumination times (2 or 12 h) were used, and results were analyzed in layers I–III and V–VI regions. At an intensity of 5 μW mm$^{-2}$, PA-Flp$_{\triangle NLS}$ efficiency was specifically increased in layer V–VI as the light illuminating duration was increased from 2 h (29.2% ± 1.5%) to 12 h (73.1% ± 3.0%). There was almost no impact of light diffusion or scattering patterns on layers I–III, whereas increasing the light intensity to 25 μW mm$^{-2}$ resulted in considerable light diffusion or scattering patterns in all layers (Supplementary Figure 6b,c).

**Data analysis**. Images from ImageXpress were analyzed using CellProfiler 2.1.0. Cells in DAPI, mCherry, or GFP channels were detected using the 'Identify Primary, Secondary or Tertiary Objects' module. The recombination percentage was calculated by detecting GFP-positive (GFP+) cells among mCherry positive (mCh +) cells using intensity threshold settings. Images from a Nikon A1 confocal microscope were analyzed using Nikon imaging software (NIS-element AR 64-bit version 4.10; Laboratory Imaging) or MetaMorph software (version 7.8.1.0, MDS Analytical Technologies). In vivo-electroporated cells, each with fluorescence signals satisfying criteria of a mean intensity greater than 1000 arbitrary units (a.u.), diameter equivalent of 10–20 μm and circularity of 0.5–1.0, were detected automatically using the 'object count' tool in Nikon imaging software. Detected cells were assumed to be marker positive (+) cells for each fluorescence channel. The

number of GFP+ (or YFP+) cells was then divided by the number of cells expressing both mCh and iRFP, and multiplied by 100 to obtain the percentage of GFP+/mCh+iRFP+ cells. The percentage of GFP+ cells among mCh+DAPI+ cells in AAV-infected tissues was calculated from automatic cell counts obtained using the 'Multi Wavelength Cell Scoring' tool in MetaMorph software. Measurements of the mean intensity (native GFP or TdTomato fluorescence) of individual cells in single confocal slices were taken from selected cell bodies or DAPI+ areas using 'Annotations and Measurements' or 'object count' tools in Nikon imaging software. A $P$-value $< 0.05$ was considered statistically significant.

**Object-exploration task**. The object-exploration task, used to detect differences in responses toward novel stimuli (novel objects) in a familiar arena, was performed according to previously described procedures[31,36] with minor modifications. Briefly, each mouse was habituated to a $20 \times 32 \times 14.5$ cm experimental cage (identical to the home cage) containing bedding for 1 h. Locomotor activity during habituation was measured using Ethovision 3.1 software (Noldus Information Technology). After habituation, each mouse was removed from the experimental cage and briefly placed in a temporary cage. Three non-identical novel objects were introduced into the experimental cage, and the mouse was immediately reintroduced into this cage. Three different shapes of wood blocks—quarter circle ($3.0 \times 3.0 \times 3.0$ cm, 15 g), cylinder (3.0 cm diameter $\times 6.0$ cm depth, 34 g), and cube ($3.0 \times 3.0 \times 3.0$ cm, 18 g)—were used as novel objects in this experiment. More than one object was used to maximize exploration. We confirmed that the mice showed no preference for any particular object. The object was placed at three different positions in the cage (Fig. 3h), and behavior was recorded for 20 min. Exploratory behavior was defined as satisfying any of the following criteria: (i) persistent projection of the nose toward the object; (ii) grabbing the object with fore limbs while keeping hind limbs fixed; (iii) touching the object using nose or whisker; or (iv) approaching such that the nose of the mouse and object were within 0.5 cm. Statistical significance was analyzed by one-way analysis of variance (ANOVA) or two-way repeated measures ANOVA followed by a Bonferroni post hoc test.

## Data availability

The data supporting the findings of this study are available within the paper and its Supplementary Information files. Extra data are available from the corresponding author upon reasonable request.

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

## Acknowledgements

We thank Daesoo Kim, Jeongjin Kim, Sangkyu Lee, Jihoon Kim, Seokhwi Kim, Minji Lee, and Na Yeon Kim for helpful discussions, and Seung-eun Lee for kindly providing the AAV-floxed-stopped-shCa$_v$3.1. This work was supported by the Institute for Basic Science (no. IBS-R001-D1), KAIST Institute for the BioCentury, and the Intelligent Synthetic Biology Center of the Global Frontier Project (2011-0031955) by the Ministry of Science, ICT and Future Planning, Republic of Korea.

## Author contributions

B.O.P., H.J., and W.D.H., conceived the idea and directed the work. H.J., B.O.P., S.-W.K., M.K., H.-S.S., and W.D.H. designed the experiments; H.J., J.H., J.K., Y.L., and B.O.P. performed plasmid constructions, AAV production, and in vivo characterization; S.-W. K. and M.K. carried out the behavior experiments; D.Y., S.K., and D.W. discussed the data with experimental support; D.Y., D.W. supported the schematic drawing of figures; H.J., S.-W.K., B.O.P., and W.D.H. wrote the manuscript.

## Additional information

**Competing interests:** The authors declare no competing interests.

