## [Peer Review File · Nature Communications]

Reviewers' comments:

Reviewer #1 (Remarks to the Author):

The manuscript by Jung et al. describes the development of a photoactivatable Flp recombinase (PA-Flp) that can be used in genome engineering applications. As the successful domain complementation in Flp was not demonstrated before (although the attempts were made more than 20 years ago when the 3D structure of Flp was not available), the authors performed an extensive analysis of such complementation based on the Flp structure. Surprisingly, the most efficient complementation was achieved by combining not the full N- and C-terminal domains of Flp but a short polypeptide from the N-terminal domain and the rest of the protein. The authors also played with the placement of the N-terminal and C-terminal domains of the Magnet system relative to the complementing Flp polypeptides to efficiently bring them into contact upon blue light illumination to assemble functional recombinase.

The authors then tested their photoactivatable Flp recombinase to activate deep mouse brain regions by illuminating them with a blue LED. In addition, the photoactivatable Flp recombinase was used to activate the Cre-lox system. The authors also documented the optimization of the duration and intensity of the blue light illumination to assemble functional Flp.

The manuscript is clearly written and easy to follow. The conclusions are supported by the data. The developed photoactivatable Flp recombinase will be definitely used in a number of genome engineering applications.

I noticed the following typo:

“Light emitting diode (LED) elimination (470±10 nm, 5-s pulse duration at 3-min intervals for 24 h)”

- should be “illumination”

Reviewer #2 (Remarks to the Author):

In this study, Jung et al create a photoactivable Flp recombinase, and apply it to enabling both local Flp recombination and non-invasive recombination in vivo. The non-invasive method was used to activate a shRNA for CaV1.3 and thereby alter behavior in an exploration protocol.

The study is an impressive technical tour de force and also creates a very useful reagent. The Flp recombinase dynamic range is impressive with very little leakiness and near-wild type levels after photoactivation. Both the local fiber-delivered and wider non-invasive activation are useful and work surprisingly well (in terms of spatial specificity and efficacy respectively). While many people would want to use Flp to achieve recombination with single-cell specificity during chronic microscopy in the brain or during development, and this was not shown here, it is certain it would work for those purposes.

I only have one moderate concern and a series of small suggestions.

Moderate concern is that in Figs 3l-m, there should be a shControl –LED group, to rule out that baseline exploration is actually lower than all the conditions tested, i.e. light alone could increase exploration and shCav3.1 leaky expression in the dark alone could increase exploration.

Minor suggestions are:

1. Optogenetic approaches should be defined in the sentence “Optogenetic approaches have the power to overcome these limitations” as the term optogenetics was first defined as controlling the activity of genetically defined neurons in the brain, but that does not appear to be what the authors mean. It is especially confusing here as the experimental system is the brain.
2. The introduction does not fairly describe the state of the art. Taslimi et al in Nature Chemical

Biology in 2016 already improved PA-Cre to PA-Cre2.0. This should be cited in the sentence “Two optogenetic modules of photoactivatable-Cre recombinase (PA-Cre) have recently been developed using split-Cre components, one in which each component is fused to CRY2 and CIB1”, and it may not be accurate to say “However, the PA-Cre/CRY2-CIB1 system shows low recombination efficiency”.

3. In addition, it is confusing to state “the applicability of the Magnet system as an efficient genetic manipulation tool in vivo has not been demonstrated”. This suggests that an important experiment to perform is to test the existing Magnet-based Cre in vivo. That Jung et al did not do this could be viewed as an key omission in this study. However I do not feel a new technology’s developer must always test all other previously existing technologies if theirs clearly has some unique advantages (in this case the ability to control recombination at FRT sites), so I am not suggesting the authors must test Magnet-based Cre in vivo. However they may not want to raise any objections about Magnet-based Cre in the introduction, since they are also using Magnets and themselves did not provide evidence that Magnet-based Cre would not work in vivo (indeed their results suggest Magnet-based Cre is likely to work in vivo).

4. The paragraph starting with “In designing PA-Flp” is too long. It could be split into several paragraphs starting with sentences “In initial screens, we constructed fusions...” and “To verify PA-Flp in vivo, we electroporated PA-Flp...”

5. The word “elimination” should be “illumination” in “In these screens, we found that blue light-emitting diode (LED) elimination...”

6. “Surprisingly, C-terminally fused split-FlpC showed dramatically increased (5- to 10-fold) recombination efficiency compared with N-terminally fused split-FlpC (Supplementary Fig. 1e)”. Supp Fig 1e does not show the comparison described.

7. “Finally, we found that, of these two split-site variants, FlpN27-nMagH with pMagH-FlpC28 (NLS-tagged) showed the highest light-dependent Flp activity (Supplementary Fig. 1e,f)”. The figures show constructs that vary only in the NLS placement, but the sentence is unclear which placement is best (and the sentence as written suggests the NLS is on the pMagH-FlpC28 which appears not to be the case). Also since PA-Flp1 is never defined in the paper, it would be appropriate to define it here. For example the sentence could be “Finally, we found that, among various combinations of Flp fragments and NLS tags, NLS-Nflp1-nMagH with pMagH-Cifp1 (designated PA-Flp1) showed the highest light-dependent Flp activity (Supplementary Fig. 1e,f)”.

8. “Ultimately, we chose PA-Flp1 (hereafter simply PA-Flp), which showed high light sensitivity ($EC_{50} = 3.1 \text{ uW/cm}^2$), fast recombination kinetics ($t_{1/2} = 1.1 \text{ h}$), and 2- to 8-fold induction compared with PA-Flp7 (Supplementary Fig. 2a, b).” As mentioned above, PA-Flp1 does not appear to be defined.

9. “To verify PA-Flp in vivo, we electroporated PA-Flp using fDIO-YFP plasmids” should be “To verify PA-Flp in vivo, we electroporated PA-Flp with fDIO-YFP plasmids”

10. “Light illumination induced a substantial increase in Flp reporter (GFP) signal compared with that observed in the dark (Fig. 1b).” could be “Light illumination at P1 induced a substantial increase in Flp reporter (GFP) signal 2 days later compared with that observed in the dark (Fig. 1b).”

11. “We also verified delivery of PA-Flp via an AAV vector in WT mice by co-infecting PA-Flp with fDIO-YFP, an exogenous Flp reporter, demonstrating a 30.4-fold increase in Flp reporter signal in the light-stimulated group compared with the non-light-stimulated group” should mention age and duration of light (I also could not find this info in the figure legend).

12. Likewise all animal figure legends should mention the age of the mice as this could affect light penetration and gene expression from AAV.

13. “We found that light illumination with a fiber-type LED ($\varnothing 0.6 \text{ cm}$) at an intensity of 50 PW/mm^2 for 30 min effectively delivered light to the cortex regions of mouse brain...” Is this more appropriately called a LED-coupled light guide?

14. Supp Fig 8c appears confusing and not necessary, one could just put an arrow to mark the location of brain dissected face in Supp Fig 8b.

15. “To test the compatibility of our system for application to loxP-flanked (floxed) lines, we investigated how to achieve photoactivatable Cre (PA-Cre) [activate Cre] using PA-Flp” would be more accurate as “To extend the applicability of PA-Flp to the many existing loxP-flanked (floxed)

lines, we investigated how to activate recombination at loxP sites using PA-Flp". As written the sentence suggests a PA-Cre was made, but one was not.

16. "Among the candidates tested, an FdCd that repositioned the NLS and Kozak sequence outside of the fDIO cassette with the mutation (Methionine to Leucine) at amino acid position 11th and 13th of Cre, which has no impact on its catalytic activity..." What do the M11L and M13L mutations do? Are they to abolish a native NLS?

17. "To evaluate applications of the PA-Flp dependent Cre (PA-FdCre) system to neurobehavioral research, we targeted mixtures of AAVs expressing PA-Flp, LF-FdCd, and Lox-shCaV3.1 (LoxP-flanked small hairpin RNA [shRNA] targeting CaV3.1) to the MS of wild-type mice (Fig. 3g)." The shRNA is not floxed, but a stop cassette upstream of it. Thus it should be floxed-stop-regulated shRNA or floxed-stopped-shRNA.

18. "Furthermore, PA-FdCre is a module of light-inducible Flp- and Cre-dual activation system, but Cre expression is featured Flp activity along with the neuron-specific promoter-directed LF-FdCd as an intersectional manner." The word "featured" should be "activated".

Point-by-point responses to reviewers' comments

We thank all the reviewers for their considered evaluation and constructive comments which were very helpful for us to improve our manuscript. We have addressed all the reviewers' points and carried out the requested experiments. Revised parts in the manuscript are designated in yellow.

Reviewer #1:

We are grateful to reviewer #1 for positive assessment of our work and the constructive comments.

Remarks to the Author:

Summary

The manuscript by Jung et al. describes the development of a photoactivatable Flp recombinase (PA-Flp) that can be used in genome engineering applications. As the successful domain complementation in Flp was not demonstrated before (although the attempts were made more than 20 years ago when the 3D structure of Flp was not available), the authors performed an extensive analysis of such complementation based on the Flp structure. Surprisingly, the most efficient complementation was achieved by combining not the full N- and C-terminal domains of Flp but a short polypeptide from the N-terminal domain and the rest of the protein. The authors also played with the placement of the N-terminal and C-terminal domains of the Magnet system relative to the complementing Flp polypeptides to efficiently bring them into contact upon blue light illumination to assemble functional recombinase.

The authors then tested their photoactivatable Flp recombinase to activate deep mouse brain regions by illuminating them with a blue LED. In addition, the photoactivatable Flp recombinase was used to activate the Cre-lox system. The authors also documented the optimization of the duration and intensity of the blue light illumination to assemble functional Flp. The manuscript is clearly written and easy to follow. The conclusions are supported by the data. The developed photoactivatable Flp recombinase will be definitely used in a number of genome engineering applications.

Specific comments:

I noticed the following typo:

“Light emitting diode (LED) elimination (470±10 nm, 5-s pulse duration at 3-min intervals for 24 h)”

- should be “illumination”

We corrected the error.

Reviewer #2:

We thank reviewer #2 for the positive assessment of our work and the constructive remarks. According to these comments, we have made following changes and improved the manuscript. Revised parts in the manuscript are designated in yellow.

Remarks to the Author:

Summary

In this study, Jung et al create a photoactivable Flp recombinase, and apply it to enabling both local Flp recombination and non-invasive recombination in vivo. The non-invasive method was used to activate a shRNA for CaV1.3 and thereby alter behavior in an exploration protocol.

The study is an impressive technical tour de force and also creates a very useful reagent. The Flp recombinase dynamic range is impressive with very little leakiness and near-wild type levels after photoactivation. Both the local fiber-delivered and wider non-invasive activation are useful and work surprisingly well (in terms of spatial specificity and efficacy respectively). While many people would want to use Flp to achieve recombination with single-cell specificity during chronic microscopy in the brain or during development, and this was not shown here, it is certain it would work for those purposes.

Specific comments:

I only have one moderate concern and a series of small suggestions.

Moderate concern is that in Figs 3l-m, there should be a shControl –LED group, to rule out that baseline exploration is actually lower than all the conditions tested, i.e. light alone could increase exploration and shCav3.1 leaky expression in the dark alone could increase exploration.

Following the reviewer's comments, we performed exploration behavioral test of shControl w/o LED group (n=5). Total object exploration time of shControl w/o LED group showed no significant difference compared with that of shControl w/ LED or shCav3.1 w/o LED group. And, we included behavioral results and detailed information of statistical analyses in the results section of our revised manuscript (Page 9 Line 12-15, Figure 3k-m).

Minor suggestions are:

1. Optogenetic approaches should be defined in the sentence “Optogenetic approaches have the power to overcome these limitations” as the term optogenetics was first defined as controlling the activity of genetically defined neurons in the brain, but that does not appear to be what the authors mean. It is especially confusing here as the experimental system is the brain.

In this sentence, we aimed to address the conceptual advantages of optogenetic approaches conferring spatiotemporal control over the conventional methods. We corrected the sentence as “Beyond these constraints, optogenetic approaches enable to control the activity of genetically defined neurons in the brain with high spatiotemporal resolution”.

2. The introduction does not fairly describe the state of the art. Taslimi et al in Nature Chemical Biology in 2016 already improved PA-Cre to PA-Cre2.0. This should be cited in the sentence “Two optogenetic modules of photoactivatable-Cre recombinase (PA-Cre) have recently been developed using split-Cre components, one in which each component is fused to CRY2 and CIB1”, and it may not be accurate to say “However, the PA-Cre/CRY2-CIB1 system shows low recombination efficiency”.

We cited the reference of PA-Cre2.0 and deleted the sentence “However, the PA-Cre/CRY2-CIB1 system shows low recombination efficiency”.

3. In addition, it is confusing to state “the applicability of the Magnet system as an efficient genetic manipulation tool in vivo has not been demonstrated”. This suggests that an important experiment to perform is to test the existing Magnet-based Cre in vivo. That Jung et al did not do this could be viewed as an key omission in this study. However I do not feel a new technology’s developer must always test all other previously existing technologies if theirs clearly has some unique advantages (in this case the ability to control recombination at FRT sites), so I am not suggesting the authors must test Magnet-based Cre in vivo. However they may not want to raise any objections about Magnet-based Cre in the introduction, since they are also using Magnets and themselves did not provide evidence that Magnet-based Cre would not work in vivo (indeed their results suggest Magnet-based Cre is likely to work in vivo).

The reviewer pointed out a key omission or question that might be raised in our study. We deleted the sentence “the applicability of the Magnet system as an efficient genetic manipulation tool in vivo has not been demonstrated”.

4. The paragraph starting with “In designing PA-Flp” is too long. It could be split into several paragraphs starting with sentences “In initial screens, we constructed fusions...” and “To verify PA-Flp in vivo, we electroporated PA-Flp...”

We splitted the paragraphs following the suggestion.

5. The word “elimination” should be “illumination” in “In these screens, we found that blue light-emitting diode (LED) elimination...”

We corrected the typing error.

6. “Surprisingly, C-terminally fused split-FlpC showed dramatically increased (5- to 10-fold) recombination efficiency compared with N-terminally fused split-FlpC (Supplementary Fig. 1e)”. Supp Fig 1e does not show the comparison described.

As the review pointed out, Supplementary Fig 1e only does not show the comparison described. Recombination efficiency of Supplementary Fig 1d should be compared with that of Supplementary Fig 1e. We have corrected as below to be easily understood.

Surprisingly, C-terminally fused split-FlpC showed dramatically increased (5- to 10-fold) recombination efficiency compared with N-terminally fused split-FlpC (see Supplementary Fig. 1e compared with Supplementary Fig. 1d).

7. “Finally, we found that, of these two split-site variants, FlpN27-nMagH with pMagH-FlpC28 (NLS-tagged) showed the highest light-dependent Flp activity (Supplementary Fig. 1e,f)”. The figures show constructs that vary only in the NLS placement, but the sentence is unclear which placement is best (and the sentence as written suggests the NLS is on the pMagH-FlpC28 which appears not to be the case). Also since PA-Flp1 is never defined in the paper, it would be appropriate to define it here. For example the sentence could be “Finally, we found that, among various combinations of Flp fragments and NLS tags, NLS-Nflp1-nMagH with pMagH-Cflp1 (designated PA-Flp1) showed the highest light-dependent Flp activity (Supplementary Fig. 1e,f)”.

Thank you for your kind suggestion with the example sentence. We have corrected the sentence as below.

Surprisingly, C-terminally fused split-FlpC showed dramatically increased (5- to 10-fold) recombination efficiency compared with N-terminally fused split-FlpC (see Supplementary Fig. 1e compared with Supplementary Fig. 1d). We termed sp1- and sp7- series with FlpNX-nMagH and pMagH-FlpCX orientation as PA-Flp1 or PA-Flp7, respectively. Additionally, we fused NLS N-terminally on PA-Flp1 or PA-Flp7 sites of FlpNX-nMagH, which showed that the potent NLS-tagging site for better efficacy is PA-Flp1 (Supplementary Fig. 1d,e). Since, the efficiency of heterodimerization between pMag and nMagH pair was comparable with pMagH and nMagH pair²³, we also tested pMag among PA-Flp1 variants. Finally, we found that among (NLS)-Nflp1-nMagH with (NLS)-pMag(H)-(NLS)-Cflp1-(NLS) tested, NLS-Nflp1-nMagH with pMagH-Cflp1 showed the highest light-dependent Flp activity (Supplementary Fig. 1e,f). (Page 4 Line 19-23, Page 5 Line 1-4).

Each Mag or MagH was defined at the first sentence of the paragraph. “In initial screens, we

constructed fusions of each of the nine split-site variants of Flp fused at their C-terminus with nMagnetHigh1 (nMagH; nMagnet variants) or pMagnetHigh1 (pMagH; pMagnet variants), which are the strongest light-dependent dimerization pair among Magnet system²³” (Page 4 Line 8-10).

8. “Ultimately, we chose PA-Flp1 (hereafter simply PA-Flp), which showed high light sensitivity ($EC_{50} = 3.1 \text{ uW/cm}^2$), fast recombination kinetics ($t_{1/2} = 1.1 \text{ h}$), and 2- to 8-fold induction compared with PA-Flp7 (Supplementary Fig. 2a, b).” As mentioned above, PA-Flp1 does not appear to be defined.

The definition of PA-Flp1 variants has been added as written in question #7. And we marked ‘PA-Flp1’ on Supplementary Fig 1e,f. We have corrected the sentence as below.

“Ultimately, we chose one of PA-Flp1 variants, NLS-Nflp1-nMagH with pMagH-Cflp1 pair (hereafter simply PA-Flp), which showed..”

9. “To verify PA-Flp in vivo, we electroporated PA-Flp using fDIO-YFP plasmids” should be “To verify PA-Flp in vivo, we electroporated PA-Flp with fDIO-YFP plasmids”

Yes. It’s right. We corrected it.

10. “Light illumination induced a substantial increase in Flp reporter (GFP) signal compared with that observed in the dark (Fig. 1b).” could be “Light illumination at P1 induced a substantial increase in Flp reporter (GFP) signal 2 days later compared with that observed in the dark (Fig. 1b).”

Yes. It’s right. We corrected it. Thank you for your example sentence.

11. “We also verified delivery of PA-Flp via an AAV vector in WT mice by co-infecting PA-Flp with fDIO-YFP, an exogenous Flp reporter, demonstrating a 30.4-fold increase in Flp reporter signal in the light-stimulated group compared with the non-light-stimulated group” should mention age and duration of light (I also could not find this info in the figure legend).

The information has been added to Figure 1 legend as below.

AAV-EF1a-PA-Flp and AAV-EF1a-fDIO-YFP were co-infected into the hippocampal DG of 8-wk-old mice, with 30 min light (0.4 mW/mm^2 , 20Hz, 20% duty cycle) or without light stimulation 2 wk after infection.

12. Likewise all animal figure legends should mention the age of the mice as this could affect light penetration and gene expression from AAV.

We have added the information of mice age on each figure legend. The information is also

included within materials and methods.

13. “We found that light illumination with a fiber-type LED (Ø 0.6 cm) at an intensity of 50 PW/mm² for 30 min effectively delivered light to the cortex regions of mouse brain...” Is this more appropriately called a LED-coupled light guide?

As the reviewer pointed out, we corrected the sentence as below.

“We found that light illumination using a light guide coupled LED (Ø 0.6 cm) at an intensity of 50 μ W/mm² for 30 min...”

14. Supp Fig 8c appears confusing and not necessary, one could just put an arrow to mark the location of brain dissected face in Supp Fig 8b.

We have deleted the previous Supplementary Fig 8c and put an arrow to mark the location of dissected surface in Supplementary Fig 8b.

15. “To test the compatibility of our system for application to loxP-flanked (floxed) lines, we investigated how to achieve photoactivatable Cre (PA-Cre) [activate Cre] using PA-Flp” would be more accurate as “To extend the applicability of PA-Flp to the many existing loxP-flanked (floxed) lines, we investigated how to activate recombination at loxP sites using PA-Flp”. As written the sentence suggests a PA-Cre was made, but one was not.

Yes. It’s right. We corrected it. Thank you for your example sentence.

16. “Among the candidates tested, an FdCd that repositioned the NLS and Kozak sequence outside of the fDIO cassette with the mutation (Methionine to Leucine) at amino acid position 11th and 13th of Cre, which has no impact on its catalytic activity...” What do the M11L and M13L mutations do? Are they to abolish a native NLS?

We admit that the manuscript is not sufficient to give information on that point which leads to confusion.

The M11L and M13L mutations of Cre has been done for deleting putative Kozak consensus sequences. A representative Kozak consensus sequence is known as “gccRccAUGG” in mammals. However, the variants acting as weak Kozak consensus sequences are known [Kozak, M. (1987). Tang et al. (2010)]. AUG is most important, because it is the actual initiation codon encoding a methionine. We hypothesized that any anti-sense transcripts containing initiation codon might act as a potential “weak kozak consensus sequence”. By introducing M11L and M13L mutations on Cre, we tried to block the translation toward anti-sense direction-encoding intact catalytic residue of Cre. In addition, we carried out an experiment examining that the mutations (M11L and M13L) has no impact on catalytic activity of Cre. Therefore, we hypothesized that the Cre leaking would be reduced by these mutations

and it seems to have an effect on reducing Cre leaking, but not completely.

Also, we have corrected the manuscript as below. The citations [Kozak, M. (1987). Tang et al. (2010)] have been added.

“To solve this problem, we designed a new FdCd candidate to block translation of ITR/LTR promoter-driven Cre (Supplementary Fig. 11a). A strategy repositioning the NLS and Kozak sequence outside of the upstream of fDIO cassette dramatically decreased the expression of Cre derived from antisense transcripts (Supplementary Fig. 11a-e). Through this strategy, antisense transcripts lost both strong kozak and functional NLS sequences of Cre in basal state, but sense transcripts (target promoter-driven NLS-tagged Cre)-mediated expression is fully induced after Flp activation. Additionally, we introduced point mutations (Methionine to Leucine) on putative Kozak consensus sequences at amino acid position 11th and 13th of Cre, which has no impact on its catalytic activity. The final version which repositioned the NLS and Kozak sequences with the dual mutations of Cre showed the least basal Cre activity among tested candidates in cultured neurons (Supplementary Fig. 11b,c) or in vivo brain (Supplementary Fig. 11d,e)...” (Page 8 Line 4-13).

17. “To evaluate applications of the PA-Flp dependent Cre (PA-FdCre) system to neurobehavioral research, we targeted mixtures of AAVs expressing PA-Flp, LF-FdCd, and Lox-shCaV3.1 (LoxP-flanked small hairpin RNA [shRNA] targeting CaV3.1) to the MS of wild-type mice (Fig. 3g).” The shRNA is not floxed, but a stop cassette upstream of it. Thus it should be floxed-stop-regulated shRNA or floxed-stopped-shRNA.

Yes. It’s right. We corrected it.

18. “Furthermore, PA-FdCre is a module of light-inducible Flp- and Cre-dual activation system, but Cre expression is featured Flp activity along with the neuron-specific promoter-directed LF-FdCd as an intersectional manner.” The word “featured” should be “activated”.

We corrected it.

REVIEWERS' COMMENTS:

Reviewer #2 (Remarks to the Author):

The authors have satisfactorily addressed my concerns.